

# Artificial intelligence-driven malware detection framework for internet of things environment

Shtwai Alsubai[1],[*], Ashit Kumar Dutta[2],[*], Abdullah M. Alnajim[3], Abdul rahaman Wahab Sait[4], Rashid Ayub[5], Afnan Mushabbab AlShehri[6] and Naved Ahmad[6]

[1] Prince Sattam Bin Abdulaziz University, Al-Kharj, Kingdom of Saudi Arabia
[2] Department of Computer Science and Information Technology, Almaarefa University, Riyadh, Kingdom of Saudi Arabia
[3] Department of Information Technology, College of computer, Qassim University, Buraydah, Saudi Arabia
[4] Department of Archives and Communication, King Faisal University, Al Ahsa, Hofuf, Kingdom of Saudi Arabia
[5] Department of Science Technology & Innovation Unit, King Saud University, Riyadh, Saudi Arabia
[6] Department of Computer Science and Information Systems, College of Applied Sciences, AlMaarefa University, Ad Diriyah, Riyadh, Kingdom of Saudi Arabia
* These authors contributed equally to this work.

Corresponding authors
Shtwai Alsubai,
Sa.alsubai@psau.edu.sa
Ashit Kumar Dutta,
adotta@mcst.edu.sa

## ABSTRACT

The Internet of Things (IoT) environment demands a malware detection (MD) framework for protecting sensitive data from unauthorized access. The study intends to develop an image-based MD framework. The authors apply image conversion and enhancement techniques to convert malware binaries into RGB images. You only look once (Yolo V7) is employed for extracting the key features from the malware images. Harris Hawks optimization is used to optimize the DenseNet161 model to classify images into malware and benign. IoT malware and Virusshare datasets are utilized to evaluate the proposed framework's performance. The outcome reveals that the proposed framework outperforms the current MD framework. The framework generates the outcome at an accuracy and F1-score of 98.65 and 98.5 and 97.3 and 96.63 for IoT malware and Virusshare datasets, respectively. In addition, it achieves an area under the receiver operating characteristics and the precision-recall curve of 0.98 and 0.85 and 0.97 and 0.84 for IoT malware and Virusshare datasets, accordingly. The study's outcome reveals that the proposed framework can be deployed in the IoT environment to protect the resources.

# INTRODUCTION

The Internet of Things (IoT) connects the real and virtual worlds (*Mu et al., 2021*; *Ben Atitallah, Driss & Almomani, 2022*). New business models and global interactions emerge as people, products, technologies, and the internet become more interconnected (*Kumar, Janet & Neelakantan, 2022*). Cybercriminals are increasingly targeting IoT devices because they are easy targets for exploiting weak authentication, outdated firmware, and malware

due to the complexity of design and implementation in hardware and software (*Khan & Salah, 2018*). The vulnerabilities in IoT applications can damage the entire network. In recent years, the exponential growth of ML techniques in identifying and categorizing IoT malware effectively. IoT devices are considered heterogeneous, can be implemented on various platforms, and have a wide range of requirements (*Li et al., 2022*). IoT is subject to the risk of being attacked by malware due to the lack of a practical malware detection (MD) framework. Malware presents a significant threat to IoT security, and one of the crucial issues is detecting unknown malware. Before attempting to implement security solutions, it is essential to understand that IoT devices have significant constraints, such as limited battery retention and low computational processing capability (*Meira et al., 2022*). Research on IoT security has drawn considerable attention from academic institutions and government agencies. Several studies have identified cyber threats and proposed countermeasures (*Emil Selvan et al., 2022*).

Conventional security approaches have proven ineffective and have failed to deliver decentralized, robust security for IoT networks (*Lan et al., 2022*). Transitioning similar solutions from traditional platforms to IoT may not be cost-effective due to the limitations of IoT devices (*Kan et al., 2021*). In addition, IoT platforms allow the integration of network resources into devices not originally conceived of as part of computer networks. Consequently, employing conventional security methods is insufficient to provide IoT systems with MD capabilities (*Khan et al., 2020*). Due to limited computing resources, conventional MD approaches failed to provide adequate security to IoT devices (*Asam et al., 2021*). The more reliable and promising performance of machine learning (ML) methods has led to their widespread adoption for MD. With ML algorithms, anti-malware tools have become more effective (*Carrillo-Mondéjar, Martínez & Suarez-Tangil, 2020*). In recent times, software define networks (SDN) offers an interactive architecture for exchanging the information (*Javeed, Gao & Khan, 2021*; *Javeed et al., 2021*; *Al Razib et al., 2022*). It consists of three layers including, application, control and data. It facilitates a wide range of security for the IoT devices. The centralized architecture and intelligence enable the developers to build an effective MD framework to identify the malware.

In the context of encrypted or compromised malware, static signature-based MD algorithms have failed (*Asam et al., 2022*). Application programming interface data is the most popular feature for building MD models based on sequential behaviours. In contrast, machine activity data is the most popular feature for depicting malware functioning using continuous behaviours (*Vignau et al., 2021*). Additionally, API calls and opcodes are often employed when developers attempt to determine the common behaviours among malware groups (*Shao, Yuan & Wang, 2021*). The behavioural-based method is a potential alternative to overcome the challenges of its counterparts. However, over-reliance on behaviours causes the MD models to misclassify a task close to benign functions or mimic lawful behaviours, resulting in a high false-positive rate.

Image-based MD can overcome the challenges and limitations of the existing MD models by minimizing the data loss (*Falana et al., 2022*). Feature engineering is one of the recent developments in image-based MD. There is a strong correlation between feature extraction and the effectiveness of the MD and classification procedure (*Saxe & Berlin,*

*2016*; *Fathurrahman, Bejo & Ardiyanto, 2022*). The existing shallow neural networks and classical ML models demand a higher training duration due to fewer hidden layers (*Lirim Ashiku, 2021*). It memorizes the training data and makes it difficult to generalize to a newer environment. Deep learning (DL) methods have become increasingly applicable to identifying and analyzing threats with ever-growing malware datasets. Recent studies focus on employing convolutional neural networks (CNN) to classify malware. Deep CNNs facilitate the development of detection systems based on malware images. It enables the MD framework to identify the crucial features of malware.

The features learned at lower layers are strengthened in higher layers. These characteristics support CNNs in producing an effective outcome (*Kumar, Janet & Neelakantan, 2022*). In addition, the computational cost is minimized by limiting the size of the dataset. The grayscale values ranges from 0 to 255 and gradually shifts between the two extremes of black and white.

Furthermore, grayscale images can be created using malware binary. The properties such as texture, intensity, and wavelet can be retrieved from the resulting images (*Liu et al., 2020a*). Furthermore, recent studies believe that the RGB image can provide more information for classifying malware images. The primary difficulty of visualization methods is computing the texture similarity of a grayscale image. These methods effectively decrypt obfuscated code. However, they are computationally expensive due to the complexity of extracting texture features from malware images. Large datasets make the feature extraction methods less efficient. Malware is constantly evolving, updating, and producing new versions of itself.

Consequently, improving the performance of the MD framework with low hardware configuration and extracting relevant information from raw binary data are the primary motivational factors for this study. The study intends to develop an MD framework using the CNN model. In addition, it applies efficient image enhancement and object detection techniques to improve the proposed framework's performance.

For IoT devices, there is a requirement for intrusion detection systems that are vastly improved and highly secured. Traditional machine learning algorithms cannot identify sophisticated cyber breaches because of their static design. DL allows for conducting a more in-depth network data analysis and spotting anomalies. Recent studies reveal the crucial role of DL in processing complex images (*Falana et al., 2022*). Visualizing malware as a coloured image gives the benefit of differentiating various components of the malware binary (*Naeem et al., 2020*; *Jian et al., 2021*; *Falana et al., 2022*). Malware programmers typically modify a small section of the malware codes to develop a new mutant. Thus, visualizing malware as an image offers the benefit of differentiating different components of the malware binary. An image-based DL-driven detection method is highly scalable, flexible, and cost-effective (*Naeem et al., 2020*; *Jian et al., 2021*). It can evaluate vast amounts of data and automatically alter security systems to identify malware or security breaches with minimum processing resources.

The contributions of the study are:

i) An effective technique to generate images from malware binaries.It overcomes the challenges of the existing RGB image generation technique. In addition, it reduces the possibility of data loss during the image generation process.

ii) A feature extraction technique for extracting the key features from the malware images. It provides the critical features for the CNN models. By presenting a set of crucial features, the performance of the MD model is improved.

iii) A hybrid CNN model for detecting malware in the IoT environment. It addresses the limitations of the existing MD techniques by employing an image-based detection technique. In addition, it demands a minimum hardware and software configuration compared to the recent CNN models.

iv) The proposed model achieved a significant outcome in detecting malware in IoT environment compared to the current models in terms of accuracy, precision, recall, and F1-measure.

The remaining part of this study is organized as follows: "Literature Review" outlines the recent MD using images and binary files. The study's methodology is discussed in "Materials and Methods". "Results" and "Discussion" highlight the performance analysis of the proposed framework and compare it with the recent MD frameworks. Finally, "Conclusion" concludes this study.

## LITERATURE REVIEW

The field of image processing extensively employed CNN to generate a practical outcome (*Smmarwar, Gupta & Kumar, 2022*). The weight sharing and the convolution kernel methods were used in CNN to overcome the limitations of neural network techniques (*Asam et al., 2021*). Recently, researchers have focused on improving the malware visualization technique's performance and reducing computation cost. This section covers visualization-related studies, including malware identification using statistical similarity measures, machine learning, and deep learning. Traditional MD techniques primarily analyse harmful code properties (*Conti, Khandhar & Vinod, 2022*). These capabilities also utilize advanced machine learning-based MD techniques to identify new forms of destructive code. However, these technologies failed to detect new malware variations.

Several malware analysis visualization methods have been suggested recently. *Makandar & Patrot (2017)* developed a novel approach for detecting malware using image features. They generated two-dimensional grayscale graphics from the structure of the compressed binary executable. Based on the findings, binary texture analysis proved more precise and efficient. *Venkatraman, Alazab & Vinayakumar (2019)* proposed an image based model for detecting malware. *Vasan et al. (2020)* proposed an approach for converting raw binaries into colour images and detecting malware families. They employed data augmentation for processing the imbalanced dataset. Malimg malware and IoT-android mobile datasets were used for performance evaluation. The outcome shows that the model can identify hidden code and malware families with limited resources.

*Liu et al. (2020b)* introduced a reinforcement method that relies on ML to identify various forms of malware and its variations. *Naeem et al. (2020)* developed an MD method

**Table 1 Related works.**

| Authors (year) | Dataset | Method | Results (%) | Platform |
|---|---|---|---|---|
| Makandar & Patrot (2017) | Malimg | Support vector machine and Discrete wavelet transform (DWT) | Average accuracy: 98.2 | Non- IoT platform |
| Venkatraman, Alazab & Vinayakumar (2019) | Malimg | CNN | Average accuracy: 98.4 | Real-world application |
| Vasan et al. (2020) | Malimg and IoT Android | Gated recurrent unit and CNN | Achieved accuracy of 98.82 on Malimg and 97.35 on IoT android | IoT platform |
| Liu et al. (2020b) | Microsoft BIG | CNN and GAN | Average accuracy: 96.25 | Real-world application |
| Naeem et al. (2020) | Leopard Mobile dataset | Deep CNN model | Average accuracy: 97.5 | Industrial IoT |
| Awan et al. (2021) | Malimg | Spatial attention CNN | Average accuracy: 97.62 | IoT platform |
| Asam et al. (2021) | Malimg | Deep boosted feature space-based malware classification | Average accuracy: 98.6 | Real-world application |
| Sharma, Sharma & Kalia (2022) | Windows malware binaries | Xception CNN | Average accuracy: 97.5 | Real-world application |
| Smmarwar, Gupta & Kumar (2022) | IoT malware and Malimg | DWT, GAN and CNN | Average accuracy: 99 | IoT platform |
| Conti, Khandhar & Vinod (2022) | Windows malware binaries | Convolutional Siamese neural network | Average accuracy: 98.5 | Real-world application |
| Yadav et al. (2022) | Android malware images | SVM and Random Forest | Achieved an average accuracy of 92.9 on multi-class and 100 on binary class | Android |
| Kumar & Janet (2021) | Malimg and Microsoft BIG | VGG16, VGG19, ResNet50, and Google's inception V3 | Average accuracy: 98.92 | Real-world application |
| Obaidat et al. (2022) | Java bytecode | CNN | Accuracy: 98.4 | Real-world application |
| Falana et al. (2022) | Malevis, Malimg, and Virusshare | Deep GAN and CNN | Average accuracy: 96.77 | Real-world application |
| Chaganti, Ravi & Pham (2022) | IoT malware | Bi-directional DL model | Average accuracy: 98 | IoT platform |
| Bensaoud & Kalita (2022) | Malimg | Deep CNN | Average accuracy: 97.5 | Real-world application |

for the Industrial Internet of Things (IIoT). To track and record information about incoming and outgoing traffic, the authors developed a sniffer gateway. Awan et al. (2021) introduced an image–based malware classification. They employed the VGG-19 network to classify 25 well-known malware images. Jian et al. (2021) suggested a unique deep neural network–based visual MD methodology. They established that three-channel RGB images are superior to grayscale images for malware identification.

Similarly, Sharma, Sharma & Kalia (2022) proposed an Xception CNN-based MD framework for classifying malware images. The authors stated that the models achieve a superior outcome than the current frameworks. Yadav et al. (2022) developed a MD framework using Andriod malware images. Obaidat et al. (2022) proposed a CNN model

using Java bytecode. *Chaganti, Ravi & Pham (2022)* developed a Bi-directional DL approach for classifying the IoT malware images. *Bensaoud & Kalita (2022)* employed Malimg dataset for evaluating the CNN model. In another study, *Falana et al. (2022)* developed a technique to convert malware binaries into an image to support the process of malware classification. A slight variation in an image assists CNN models in identifying critical malware. They employed three benchmark datasets: MaleVis, Mallmg, and Virusshare. The findings suggested that the model achieves an average accuracy of 96.77%. The recent techniques focussed on pattern-based MD. However, it has many drawbacks, including a high false positive rate that causes many valid activities to be incorrectly labelled intrusive. There is a demand for more critical training data. In addition, the existing methods require high-end computation resources to generate an effective outcome. Table 1 outlines the features of the existing MD frameworks.

## MATERIALS AND METHODS

The authors propose a DL based framework to classify malware and benign images based on the study's objective. Figure 1 highlights the three phases of the proposed framework. In phase 1, the authors convert the binaries into an image. The images are pre-processed and resized as $600 \times 600$ pixels. The authors employed you look only once (Yolo) V7 to identify critical features from the images. Phase 2 involves Harris Hawks optimization (HHO) to fine-tune the DenseNet161 parameters to identify malware from the datasets. Finally, phase 3 evaluates the performance of the proposed frameworks. The authors utilize two datasets in this study, including IoT malware and binaries. Two IoT malware datasets (IoT_malware and Virusshare) are used in this study which is available in the Elmasry dataset (*Malware, 2021*) and Virusshare dataset (*Virusshare, 2021*) respectively. IoT_malware dataset is a recently developed malware images dataset. It includes the IoT malware images of categories including benign and malware. The unpacked executable and linkage format binaries for malware and benign applications were represented in the image format. In addition, the Virusshare dataset contains instances of multiple malware families. The description of the datasets is provided in Table 2.

Based on the SDN framework, the researchers framed the network model as shown in Fig. 2 for implementing the proposed model. In the control layer, the binaries are converted into images and transformed as RGB images. Yolo V7 extracts the crucial objects. Finally, the fined tuned CNN model classifies the malware and benign images.

In phase 1, the authors follow the approaches of *Falana et al. (2022)* to convert binaries into an image. Let $B_1$, $B_2$, …, $B_n$ and $M_1$, $M_2$, …, $M_n$ be the benign and malware binaries set, respectively. Let D be a space to hold benign and malware binaries. Therefore, $D_i$ represents a binary, which may be benign or malicious. Figure 3 shows converting binaries and grayscale images into RGB images.

The following algorithm presents the algorithm for transforming the binaries (D) into a grayscale image. During the image pre-processing phase, the grayscale images (G) are converted into RGB images (RGB). Initially, the luminosity method converts a grayscale image into an RGB image. Equation (1) represents the conversion process of G into RGB.

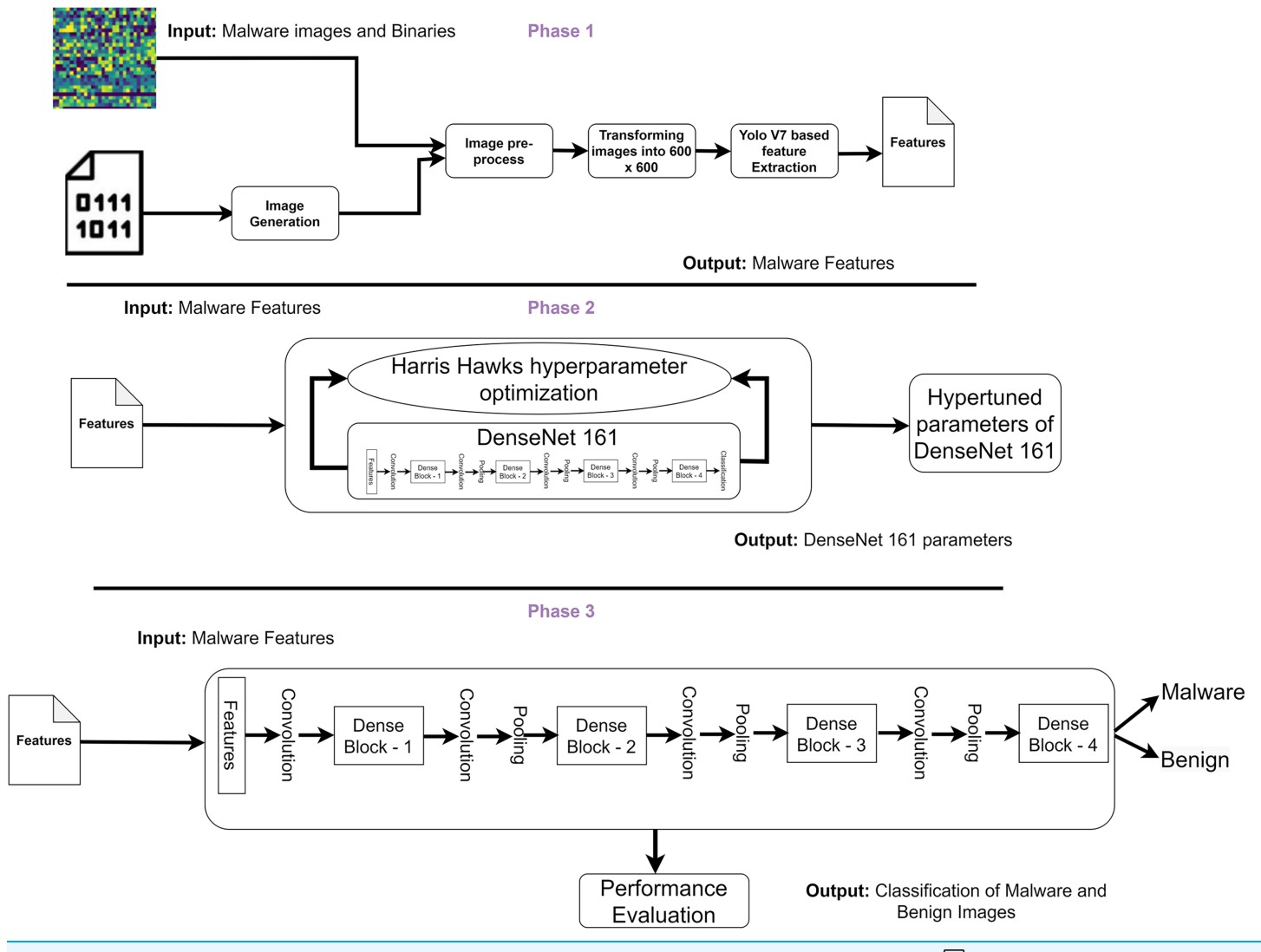

**Figure 1 Proposed malware detection framework.** 

| Table 2 Dataset characteristics. | | | |
|---|---|---|---|
| **Dataset** | **Files** | **Classification** | **Size (in MB)** |
| IoT_malware | 17,186 | 2 | 64 |
| Virusshare | 30,967 | 2 | 73 |

$$\text{RGB}(I) = G[(0.3 * R) + (0.59 * G) + (0.11 * B)] \tag{1}$$

Insufficient or non-uniform RGB images contain a significant amount of noise. Therefore, the authors enhance the RGB image based on *Mu et al. (2021)*. Equation (2) shows the intensity enhancement of RGB(x,y).

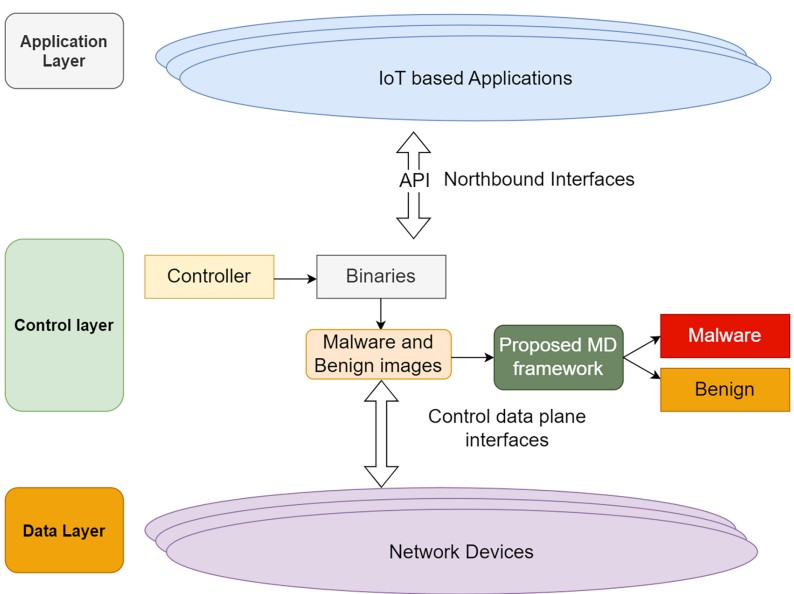

**Figure 2  Proposed network model.**

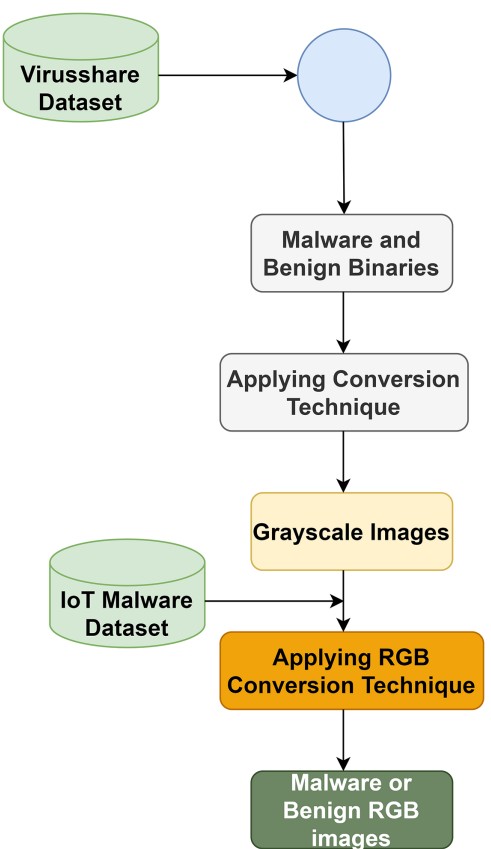

**Figure 3  RGB image conversion process.**

---

| **Algorithm** Grayscale to RGB image. |
|---|
| Input: D |
| Output: Grayscale Image (I) |
| Bit_array = 0; // 8 bit array |
| Pix_arr[0][0] = 0; // A pixel array |
| I_width = $2^8$ |
| I_height = size of (D)/$2^{11}$ |
| for i = 1 to len (D) – 1 do |
|      for c = 1 to F |
|      if c = 0 or c=1 |
|        Bit_array = c |
|      if I % 8 ==0 |
|        Continue |
|      endif |
|      endif |
|       endfor |
|     for k = 1 to 255 |
|      X[k] = Bit_array[k] |
|      for l = 1 to len(Bit_array) |
|       Pix_arr[k][l] = Bit_array[k] |
|        endfor |
|      endfor |
|   endfor |
|  return pix_arr |

$$RGB(x, y) = a_k RGB(x,y) + \overline{b_k} \tag{2}$$

where $a_k$ and $b_k$ are linear coefficients.

Equations (3)–(5) outline the process of brightness equalization using adaptive gamma correction.

$$RGB(x,y) = RGB(x,y)^{\varnothing(x,y)} \tag{3}$$

where $\varnothing(x,y)$ is the gamma correction.

$$\varnothing(x,y) = \frac{RGB(x,y) + \propto}{1 + \propto} \tag{4}$$

$$\propto = 1 - \frac{1}{cd} \sum_{x=1}^{c} \sum_{y=1}^{d} RGB(x,y) \tag{5}$$

where $\propto$ is the adaptively derived mean value of G(x,y), and c, d is the height and width of RGB(x,y).

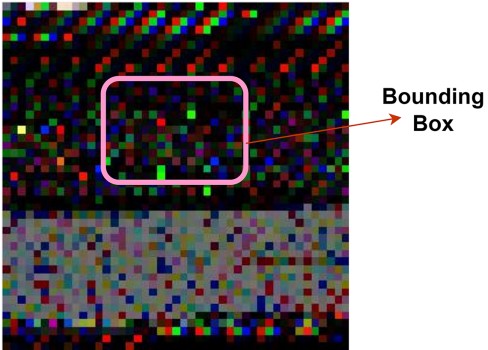

**Figure 4  Bounding box region.**           

In addition, image fusion filters noises from the RGB image. Equations (6)–(8) show the denoised reflection component (DNI) and enhanced RGB image.

$$DNI = \overline{a_i}RGB(x,y) + \overline{b_i} \tag{6}$$

$$Enhanced\ RGB(x,y) = \frac{1}{1 + \exp\left(-8(DNI - b)\right)} \tag{7}$$

$$b = \frac{1}{cd}\sum\nolimits_{x=1}^{c}\sum\nolimits_{y=1}^{d} RGB(x,y) \tag{8}$$

where b is image intensity, and c and d are the images' height and width.

In this phase, the RGB images are enhanced in order to assist the DenseNet model. The high—standard deviation in the images are adjusted to reduce the variation in the pixel value. Image fusion filter is applied to remove the noises by blurring the images. It adjusts the uneven pixel value and removes the chromatic aberration. In the subsequent step, Canny edge detection is employed for identifying the ranges of edges. The non-maximum suppression is used to thin out the edges. The intensity of the images are identified using Double threshold method. Finally, contrast limited adaptive histogram equalization is employed to improve the image quality.

Furthermore, the authors employ Yolo V7 (*Wang, Bochkovskiy & Liao, 2022*) to extract meaningful features from the images. Yolo V7 achieves a superior outcome with fewer computational resources. It generates an output faster without any pre-trained weights. It uses CNN for extracting features and predicting the probability of classes. Yolo V7 overcome the challenges in its previous versions. It contains an extended efficient layer aggregation network and compound model scaling technique that support the proposed model for detecting malware images. Yolo V7 architecture includes residual blocks, bounding box and intersection over union (IoU). It divides the images into multiple grids (residual blocks) with equal dimensions. Each grid is a region to highlight the object. It consists of width (w), height (h), class (c), and center (x,y).

Equation (9) represents the bounding box that highlights a region in Fig. 4.

$$Y = (P_c, bb_x, bb_y, bb_h,\ bb_w, C) \tag{9}$$

where $P_c$ is the probability of an object in the bounding box (bb).

IoU is a metric for evaluating the performance of Yolo V7. It measures the Yolo V7's ability to detect the malware dataset's features. Equation (10) shows the expression of IoU.

$$IoU = \frac{Area\ of\ overlapping\ of\ actual\ and\ predicted\ malware\ feature}{Area\ of\ union\ of\ actual\ and\ predicted\ malware\ feature} \tag{10}$$

Yolo V7 computes the IoU score for each object detection process. An IoU score greater than 0.5 represents the better performance of an object detection model.

In phase 2, the authors optimize the DenseNet161 model using the HHO algorithm. HHO algorithm is one of the recent optimization algorithms for improving the performance of the complex models. It tunes the CNN model's parameters for improving the classification accuracy. It minimizes the error rate and searches for the optimal learning rate for identifying the malware and benign images. The architecture of DenseNet161 comprises an activation function, a pooling layer, a dropout layer, and the convolutional layer. Each layer acquires information from the previous layer and guides the subsequent layers. DenseNet161 simplifies the connecting pattern among the layers. It reuses malware and benign image features and enhances the network's performance. In addition, it requires a limited number of parameters compared to its counterparts. The developmental rate controls the number of data in a layer. Each dense block includes two convolutions, and each dense layer contains two operations to extract malware and benign features and reduces its depth. HHO is a familiar swarm-based optimization technique. It is used to improve the performance of the DenseNet161 model. In the context of MD, HHO identifies the effective parameters (number of pooling layers, dropout layer, and convolutional layers) for generating the outcome.

The malware and benign images are considered a rabbit in the HHO searching environment. The HHO searching strategies support the proposed framework to classify the images. The exploration and exploitation phases assist the DenseNet161 model in identifying the malware's exact location and benign images. Let q be the equal chance between the DenseNet161 parameters. The HHO exploitation phase for the proposed framework is modelled in Eq. (11).

$$M(t+1) = \begin{cases} (M_{rand}(t) - Y_1|M_{rand}(t) - 2Y_2M(t)), & q \geq 0.5 \\ M_{malben}(t) - M_m(t) - Y_3(LB + Y_4(UB - LB)), & q < 0.5 \end{cases} \tag{11}$$

where M(t+1) is the location of the DenseNet161 parameters in the subsequent iteration, $M_{malben}$ (t) is the location of malware and benign image. M(t) is the present position of hawks, Y1, Y2, Y3, Y4, and q are arbitrary numbers between 0 and 1, frequently modified at each iteration. LB and UB are the lower and upper bounds of each variable, $M_{rand}(t)$ is an arbitrary DenseNet161 parameter from the present population, and $M_m$ is the average location of the parameter. Equations (12) and (13) represent the soft besiege of malware and benign images in the HHO environment.

$$M(t+1) = \Delta Mt) - E|JM_{malben}(t) - M(t)| \tag{12}$$
$$\Delta M(t) = M_{malben}(t) - M(t) \tag{13}$$

where $\Delta M(t)$ is the difference between the malware and benign image and the present

position in iteration t, E is the parameter to represent the transition between soft and hard besiege, and J is the jump strength. Hard besiege is described in Eq. (14).

$$M(t + 1) = M_{malben}(t) - E|\Delta M(t)| \tag{14}$$

The following algorithm outlines the HHO algorithm for optimizing the Densenet161 parameters.

In phase 3, the authors apply precision, recall, F1-measure, accuracy, Matthews correlation coefficient (MCC), and Kappa to evaluate the proposed framework's performance. The dataset is divided into a train set (70%) and test set (30%). In the MD environment, precision is the number of malware and benign classification among the classified images. A recall is a set of classified malware and benign images. F1-score is the harmonic mean of a number of malware and benign images in the datasets and correctly detected images. Accuracy is the number of optimally classified malware and benign images. MCC is the difference between predicted malware and benign images and actual malware and benign images.

Furthermore, it summarizes the confusion and error matrices. Cohen's Kappa compares the classified malware and benign images with the expected accuracy. It addresses the evaluation bias by providing the chances of generating optimal classification using a random guess. In addition, the error rate and computation cost are calculated for each classification.

## RESULTS

In this section, the authors highlight the experimental outcome of this study. The proposed model is implemented in Windows 10 professional environment, i7 processor, GTX 1080 Ti (11 GB). Python 3.9 with Keras (*Keras, 2022*) library is employed for developing the proposed framework, *Vasan et al. (2020)* framework, *Jian et al. (2021)* framework, *Sharma, Sharma & Kalia (2022)* framework, and *Falana et al. (2022)* framework. In addition, YoloV7 (*Wang, Bochkovskiy & Liao, 2022*), DenseNet161 (*DenseNet161, 2022*) and HHO (*HHO, 2022*) are utilized for constructing the model.

The similar hardware and software configuration is followed for the training phase. During the training phase, the DenseNet161 parameters are supervised by the HHO algorithm. The authors train the DenseNet161 model with IoT datasets under the HHO environment to identify critical parameters for generating an optimal outcome. During the training phase, the proposed MD framework generates an optimal result at the 32[nd] and 37[th] epoch for IoT malware and Virusshare datasets, respectively. Furthermore, the authors extended the training to the 37[th] and 40[th] epochs for the IoT malware and Virusshare datasets. However, there is no significant improvement in the model's performance. Thus, epoch values and the dropout ratios of 32 and 41, 0.3 and 0.5, are assigned for IoT malware and Virusshare datasets, respectively. Based on the outcome of the hyperparameter optimization, an array of five layers comprised of two fully connected layers, three dropout layers and an activation function are integrated with the DenseNet161 model. The hyperparameter tuning process identifies an optimal set of DenseNet161 parameters to detect malware images from the dataset.

| Algorithm HHO pseudocode for DenseNet161. |
|---|

Input: Dataset D and number of iteration (epoch) T

output: Malware image classification and its fitness value

Initialize the population at random $M_i$ (i = 1, 2,…, D)

While (d in D) do

    Calculate the fitness value (parameters) of DenseNet161

    Set $M_{malben}$ as the location of malware and benign images

For each parameter ($M_i$) do

Update $E_0$ and J // $E_0$ = 2 rand () −1 and J = 2(1-rand ())

update E using $E = 2E_0\left(\dfrac{T-t}{T}\right)$

  if ($|E| \geq 1$) then

    Update the location using Eq. (11)

  if ($|E| < 1$) then

      if($r \geq 0.5$ &$|E| \geq 0.5$) then

        Update the location using Eq. (12)

      else if ($r \geq 0.5$ &$|E| < 0.5$) then

        update the location using Eq. (14)

return $M_{malben}$

**Table 3 Performance analysis for the IoT malware dataset.**

| Methods/Measures | Accuracy | Precision | Recall | F1-measure | MCC | Kappa |
|---|---|---|---|---|---|---|
| Training | | | | | | |
| Malware | 98.3 | 98.1 | 98.3 | 98.2 | 97.8 | 97.9 |
| Benign | 98.6 | 98.4 | 98.5 | 98.45 | 97.5 | 98.1 |
| Average | 98.45 | 98.25 | 98.4 | 98.33 | 97.65 | 98 |
| Testing | | | | | | |
| Malware | 98.7 | 98.6 | 98.1 | 98.35 | 97.6 | 97.5 |
| Benign | 98.6 | 98.8 | 98.5 | 98.65 | 97.4 | 97.8 |
| Average | 98.65 | 98.7 | 98.3 | 98.5 | 97.5 | 97.65 |

Table 3 outlines the performance of the proposed framework. In the testing phase, the trained DenseNet161 model achieves an average accuracy, precision, recall, F1-measure, MCC, and Kappa of 98.65, 98.7, 98.3, 98.5, 97.5, and 97.65, respectively, for the IoT malware dataset. HHO assists the DenseNet161 model in generating optimum results. The outcome reveals the adequate performance of the proposed MD model. The higher value of MCC and Kappa indicates that the proposed model classifies the images with optimal precision on the imbalanced dataset.

    Likewise, Table 4 shows the proposed framework's performance on the Virusshare dataset. Compared to the IoT malware dataset, the Virusshare dataset contains many files.

| Table 4 Performance analysis for the Virusshare dataset. | | | | | | |
|---|---|---|---|---|---|---|
| Methods/Measures | Accuracy | Precision | Recall | F1-measure | MCC | Kappa |
| Training | | | | | | |
| Malware | 97.5 | 97.8 | 96.3 | 97.04 | 95.8 | 95.4 |
| Benign | 97.1 | 96.5 | 97.1 | 96.8 | 95.7 | 95.6 |
| Average | 97.3 | 97.15 | 96.7 | 96.92 | 95.75 | 95.5 |
| Testing | | | | | | |
| Malware | 97.4 | 96.1 | 97.2 | 96.65 | 95.2 | 94.8 |
| Benign | 97.2 | 96.7 | 96.5 | 96.6 | 95.3 | 94.9 |
| Average | 97.3 | 96.4 | 96.85 | 96.63 | 95.25 | 94.85 |

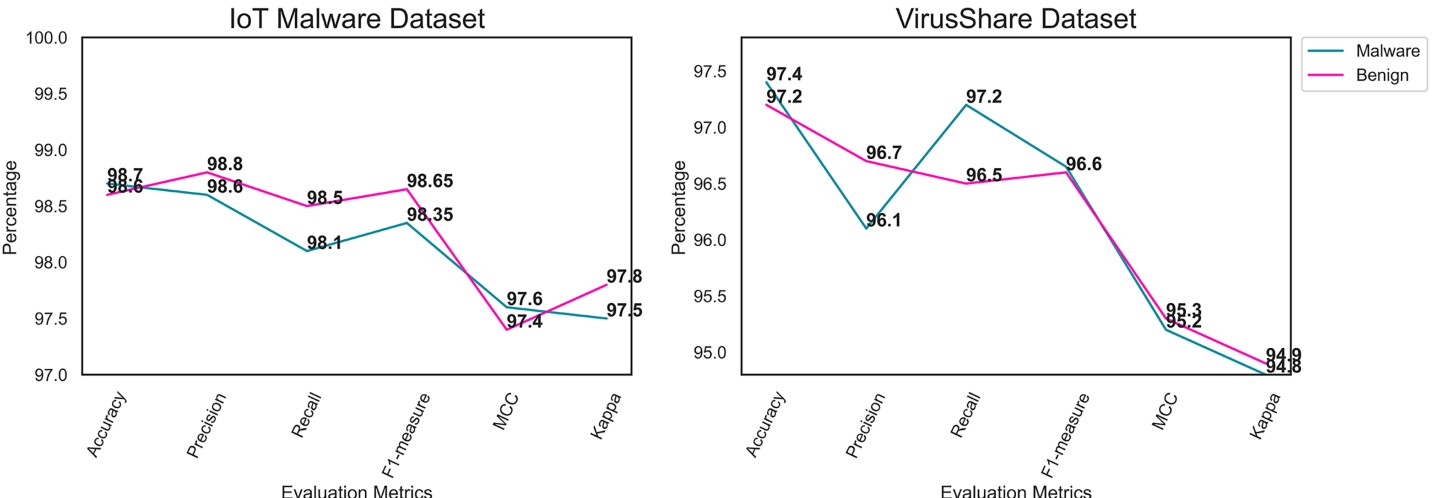

**Figure 5 Performance analysis outcome of the proposed framework for IoT malware and Virusshare Datasets.**

Moreover, the image conversion model supports the proposed framework for converting the binaries into an RGB image. The proposed model achieves an optimal accuracy on the Virusshare dataset. The feature extraction process assists the proposed model in identifying the crucial features of the images. Figures 5A and 6B illustrate the performance of the proposed framework on the IoT malware and Virushare datasets. It shows that the model effectively classifies the malware and benign images. In addition, the proposed model addresses the overfitting challenges on the IoT malware and the Virusshare datasets.

Table 5 highlights the comparative analysis's outcome of the MD framework. The proposed framework outperforms the recent MD frameworks. The high value of Kappa suggests the effectiveness of the proposed MD framework on the imbalanced dataset. In addition, it highlights the importance of the proposed MD framework in handling true and false positives. However, *Falana et al. (2022)* framework produces a reasonable outcome on the IoT malware dataset.

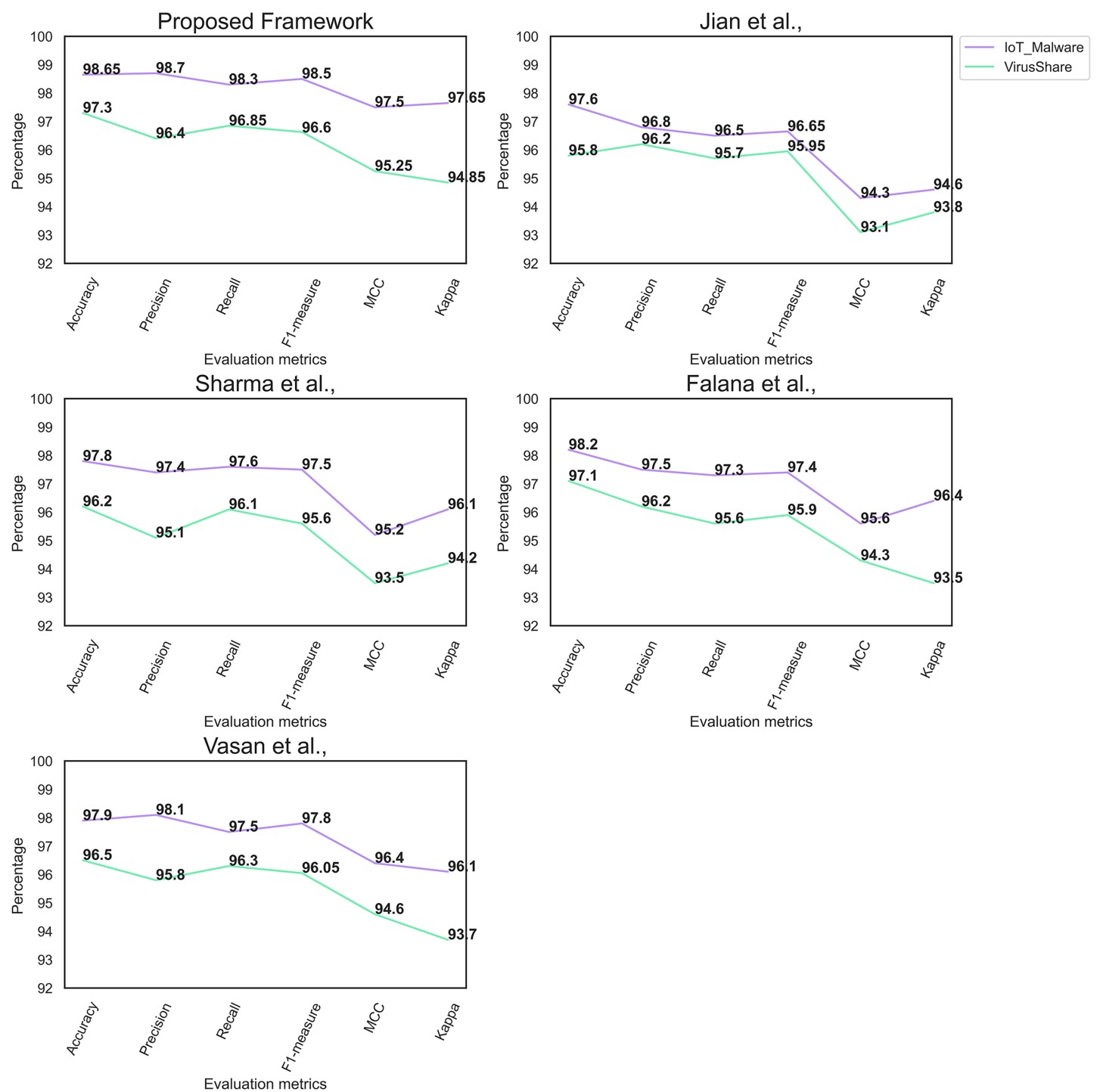

**Figure 6  Comparative analysis outcome of malware detection frameworks.**

**Table 5  Comparative analysis of the IoT malware dataset.**

| Methods/Measures | Accuracy | Precision | Recall | F1-measure | MCC | Kappa |
|---|---|---|---|---|---|---|
| Proposed framework | 98.65 | 98.7 | 98.3 | 98.5 | 97.5 | 97.65 |
| *Jian et al. (2021)* framework | 97.6 | 96.8 | 96.5 | 96.65 | 94.3 | 94.6 |
| *Sharma, Sharma & Kalia (2022)* framework | 97.8 | 97.4 | 97.6 | 97.5 | 95.2 | 96.1 |
| *Falana et al. (2022)* framework | 98.2 | 97.5 | 97.3 | 97.4 | 95.6 | 96.4 |
| *Vasan et al. (2020)* framework | 97.9 | 98.1 | 97.5 | 97.8 | 96.4 | 96.1 |

**Table 6  Comparative analysis of the Virusshare dataset.**

| Methods/Measures | Accuracy | Precision | Recall | F1-measure | MCC | Kappa |
|---|---|---|---|---|---|---|
| Proposed framework | 97.3 | 96.4 | 96.85 | 96.63 | 95.25 | 94.85 |
| *Jian et al. (2021)* framework | 95.8 | 96.2 | 95.7 | 95.95 | 93.1 | 93.8 |
| *Sharma, Sharma & Kalia (2022)* framework | 96.2 | 95.1 | 96.1 | 95.6 | 93.5 | 94.2 |
| *Falana et al. (2022)* framework | 97.1 | 96.2 | 95.6 | 95.9 | 94.3 | 93.5 |
| *Vasan et al. (2020)* framework | 96.5 | 95.8 | 96.3 | 96.05 | 94.6 | 93.7 |

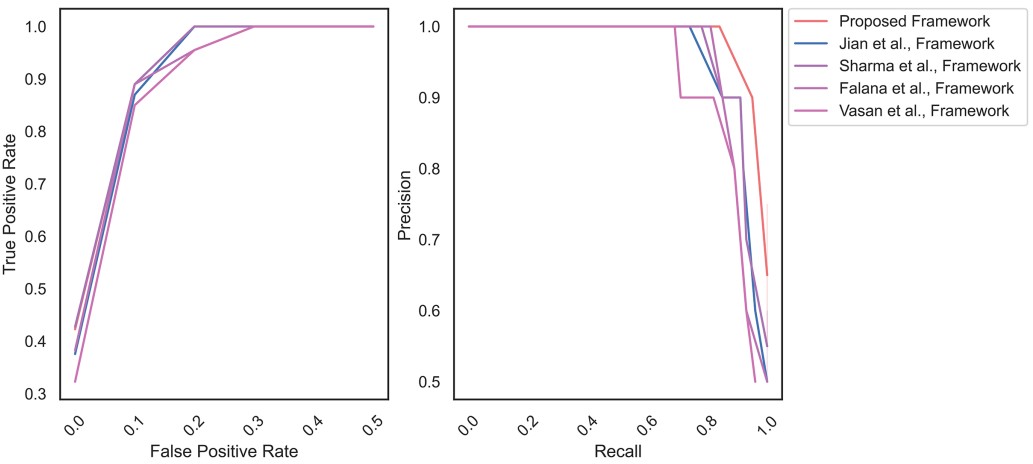

**Figure 7  AU-ROC and AU-PRC for IoT malware dataset.**

Likewise, Table 6 presents the results of the comparative analysis of the Virusshare dataset. The proposed MD framework obtained a superior MCC and Kappa on the Virusshare dataset. Yolo V7 and HHO algorithm enables the proposed framework to produce a superior outcome. In addition, the image enhancement technique offers the proposed framework to identify the key objects. However, both *Falana et al. (2022)* and *Vasan et al. (2020)* frameworks achieve results similar to the proposed framework.

Figure 6 reflects the performance of the individual MD frameworks on the IoT malware and the Virusshare datasets, respectively. The proposed feature extraction method offers

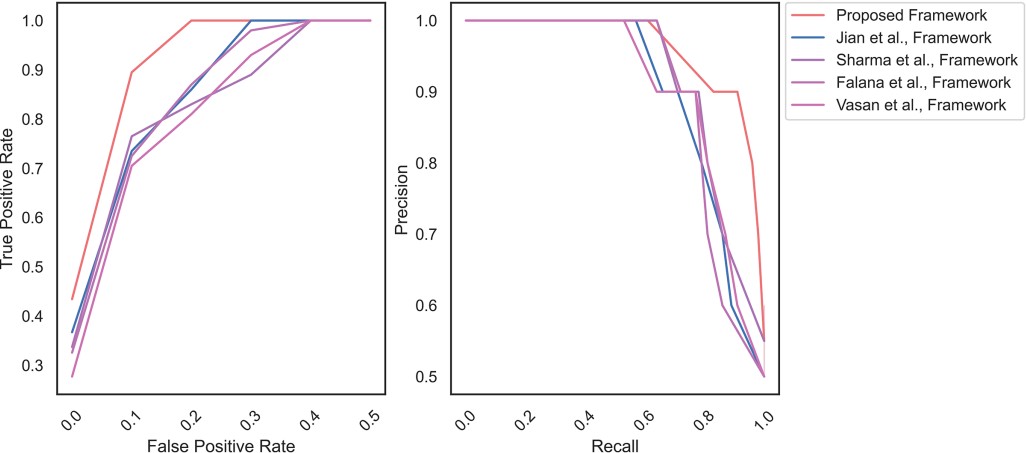

**Figure 8  AU-ROC and AU-PRC for Virusshare dataset.**

**Table 7  Error rate of CNN models.**

| Methods/Dataset | IoT malware dataset (%) | Virusshare dataset (%) |
|---|---|---|
| Proposed framework | 14.2 | 15.6 |
| *Jian et al. (2021)* framework | 14.8 | 16.4 |
| *Sharma, Sharma & Kalia (2022)* framework | 15.3 | 17.1 |
| *Falana et al. (2022)* framework | 15.7 | 16.2 |
| *Vasan et al. (2020)* framework | 16.3 | 15.9 |

the key features of the DenseNet161 model to classify malware and benign images effectively.

Figure 7 highlights the area under the Receiver operating characteristics (AU-ROC) and the MD frameworks' precision-recall curve (AU-PRC) on the IoT malware dataset. The proposed framework achieves AU-ROC and AU-PRC values of 0.98 and 0.85, respectively. On the other hand, the AU-ROC and AU-PRC values of *Jian et al. (2021)* (0.88 and 0.75), *Sharma, Sharma & Kalia (2022)* (0.89 and 0.77), *Falana et al. (2022)* (0.87 and 0.69) and *Vasan et al. (2020)* (0.84 and 0.74), accordingly. The outcome represents the classification efficiency of the proposed MD. The higher values of AU-ROC and AU-PRC indicates that the MD model detects the features of malware and benign images, effectively.

Likewise, Fig. 8 displays the AU-ROC and AU-PRC of the MD frameworks. The proposed framework reaches the AU-ROC and AU-PRC of 0.97 and 0.84, respectively. In contrast, the remaining frameworks achieve the AU-ROC and AU-PRC of *Jian et al. (2021)* (0.92 and 0.77), *Sharma, Sharma & Kalia (2022)* (0.90 and 0.81), *Falana et al. (2022)* (0.92 and 0.73), and *Vasan et al. (2020)* (0.78 and 0.76), accordingly.

Table 7 outlines the error rate of MD frameworks. The proposed framework produces fewer errors for IoT malware (14.2%) and Virusshare (15.6%). The feature extraction phase assists the proposed framework in generating a superior outcome compared to the other frameworks.

**Table 8 Computational complexities of the CNN models.**

| Methods/Dataset | IoT malware dataset | | | Virusshare dataset | | |
|---|---|---|---|---|---|---|
| | No. of parameters | Learning rate | Testing time (seconds) | No. of parameters | Learning rate | Testing time (seconds) |
| Proposed framework | 6.4 M | $1 \times 10^{-4}$ | 152.3 | 7.1 M | $1 \times 10^{-5}$ | 148.9 |
| Jian et al. (2021) framework | 6.8 M | $1 \times 10^{-3}$ | 167.2 | 8.4 M | $1 \times 10^{-4}$ | 153.4 |
| Sharma, Sharma & Kalia (2022) framework | 7.4 M | $1 \times 10^{-2}$ | 154.5 | 8.7 M | $1 \times 10^{-3}$ | 161.6 |
| Falana et al. (2022) framework | 7.3 M | $1 \times 10^{-3}$ | 151.6 | 7.6 M | $1 \times 10^{-4}$ | 159.5 |
| Vasan et al. (2020) framework | 6.5 M | $1 \times 10^{-3}$ | 150.5 | 6.9 M | $1 \times 10^{-5}$ | 161.1 |

Finally, Table 8 presents the computational complexities of the MD frameworks in classifying the malware images. Compared to the existing frameworks, the proposed framework consumes fewer parameters, learning rate, and computation time.

## DISCUSSION

The authors developed an image-based MD framework for identifying malware and benign files in the IoT environment. An image conversion technique converts malware and benign binaries into a grayscale image. Furthermore, the grayscale images are enhanced to RGB images. An object identification technique extracts a key feature from the images. Yolo V7 is a recent CNN technique for identifying the crucial elements of malware and benign images. HHO algorithm is used to optimize the DenseNet161 model for classifying malware and benign images. It identifies the critical parameters of the DenseNet model in order to detect malware within a limited amount of time. DenseNet161 contains a set of hyper-parameters that reinforces the model to find the crucial objects from the images. Predictive accuracy and detection rates are the primary metrics for evaluating MD frameworks. The primary step in securing a system and gaining control over its further malware spread is accurately discovering the previously undetected instances. Improving the detection accuracy of a proposed method may result in false alarms. Attempts to reduce false alarms may have an unintended negative effect on detection efficiency. As a result, the proposed model uses dissimilarity by contrasting the harmonic mean of both factors, known as the F1 measure. In addition, MCC and Kappa are used to measure the efficiency of the proposed framework.

The image format enables the MD framework to serve multiple types of platforms. In addition, the CNN model can identify a slight variation in textures and patterns in the images. Thus, the proposed model supports the SDN framework to offer a protective environment for the IoT devices. The study uniquely integrates image enhancement, object detection (Yolo V7), and hyper-parameter tuned CNN model (HHO—DenseNet161). Image enhancement and object detection reduces the computation overhead of the proposed model. The hyperparameter optimization tunes the key parameters such as number of dropout layers and epochs. The fined tuned model classifies the images with limited resources.

*Jian et al. (2021)* developed a deep neural network architecture called SERLA (SEResNet50 + Bidirectional Long Short Term Memory (Bi-LSTM) + attention) to increase the performance of the detection approach. However, the proposed model's performance outperforms the Jian et al., model due to the effective image enhancement and feature extraction techniques. In addition, the computation cost for constructing Bi-LSTM is higher than the proposed method. The framework of *Sharma, Sharma & Kalia (2022)* generated a better outcome; however, the computation cost is higher than the proposed MD framework. *Falana et al. (2022)* framework comprised a CNN and generative neural network for classifying the malware images. However, there is a lack of feature engineering or extraction process to identify the critical features from the images. In addition, the complex architecture requires additional computation time to generate the outcome. In line with the *Vasan et al. (2020)* framework, the recommended MD framework applied the HHO algorithm to fine-tune the DenseNet161. Figure 6 reflects the MD performance on IoT malware and Virusshare datasets. It shows that the proposed MD outperforms the recently developed image-based MD. In line with the studies (*Obaidat et al., 2022*; *Yadav et al., 2022*; *Smmarwar, Gupta & Kumar, 2022*), the proposed model achieves a superior outcome. The significant improvement in the feature extraction and image classification processes enabled the proposed MD to achieve a better outcome. The existing models (*Chaganti, Ravi & Pham, 2022*; *Kumar, Janet & Neelakantan, 2022*) generated a reasonable outcome. However, the computation cost was very high comparing to the proposed MD framework. Tables 7 and 8 reveals the error rates and the computational complexities of the MD frameworks. It is evident that the proposed model require less computational resources for detecting the malware. The proposed study's outcome follows the studies of *Vinayakumar et al. (2018)*, *Su et al. (2018)*, *Rabbani et al. (2020)*, *Naeem et al. (2020)*, *Javeed, Gao & Khan (2021)*, *Javeed et al. (2021)*, *Anand et al. (2021)*, *Awan et al. (2021)* and *Al Razib et al. (2022)* for protecting the computational resources from the malware.

The presently offered MD technologies are only effective on traditional networks. The implementation of the models are difficult to apply on IoT networks or do not possess the flexibility and robustness necessary to ensure secure operations. The study's outcome reveals that they are appropriate for securing the IoT. It is adaptable, distributed, resilient, and does not require many computational resources. Many IoT devices, including temperature and humidity sensors, used in environmental and agricultural applications are battery-powered and deployed in distant places, necessitating an MD technique that is both computationally and energy-efficient to extend the battery life of these devices. The proposed framework can be applied in environmental and agricultural applications to minimize energy consumption and protect the network. IoT-based systems in smart cities rely on various devices, such as security cameras, that collect personal information and need stringent security protocols to prevent unauthorized access. Safeguarding the IoT system against malware is critical for the well-being of the workforce and the sustained

improvement of the Industrial IoT. Thus, the proposed MD framework can offer an effective industrial working environment and safeguard crucial computing resources in industrial settings.

The proposed model yields reliable results and aids in identifying malware in IoT networks. In future investigations, several limitations should be addressed. CNN's multiple layers increase training time and demand a GPU. Nevertheless, the current IoT framework facilitates the high end software and hardware configuration for implementing a DL based detection method. In addition, the proposed MD model is a lightweight application comparing to the recent models. Therefore, the proposed MD model can operate in multiple IoT platforms. The study's findings reveal that the proposed MD model require limited computational resources. The hyperparameter tuned CNN model achieved a better outcome. The existing CNN and recurrent neural network approaches failed to present a crucial pattern from the malware binaries due to data loss and irrelevant features. Yolo V7 model assists the proposed MD framework by providing the key features of malware. The proposed image based MD framework overcome the challenges of the existing approaches.

The proposed technique may suffer from the imbalanced dataset. The data pre-processing is required to improve an image's quality and deliver high performance. There is a possibility of losing critical features due to multiple features. The inability to use coordinate frames might render the graphics unfavorably. The architecture of the proposed model necessitates a sizable quantity of data to yield an exciting result. However, the researcher introduced image enhancement and feature extraction to handle the shortcomings of the CNN model. Incorporating feature selection results into the images' internal representation can yield positive results.

## CONCLUSION

The authors present the image-based MD framework for the IoT environment in this study. The malware binaries are converted into images to improve the quality of the malware classification approach. In addition, an image enhancement technique is employed to convert the grayscale images to RGB images. An object identification method is used for feature extraction to support the trained convolutional neural network approach. For classifying the malware images, the authors employed the DenseNet161 model with the support of the Harris Hawks optimization algorithm. The performance evaluation was conducted on IoT malware and Virusshare datasets. The experimental outcome shows that the proposed framework is suitable for real-time applications.

Moreover, the framework is lightweight, which demands a low computation cost for generating effective results. Thus, the framework can be applied to small and large-scale industries. It performs better on IoT malware and Virusshare datasets. However, there is a demand for additional experimentation to improve the performance of the proposed MD framework. In the future, the authors intend to extend the framework with the generative adversarial network to generalize the proposed framework's implementation to other malware image datasets.

### Funding

The authors received support from AlMaarefa University while conducting this research work. This study is supported via funding from Prince Sattam bin Abdulaziz University project number (PSAU/2023/R/1444). This work was supported by the Deanship of Scientific Research, Vice Presidency for Graduate Studies and Scientific Research, King Faisal University, Saudi Arabia [Grant No. 2740]. The funders had no role in study design, data collection and analysis, decision to publish, or preparation of the manuscript.

### Grant Disclosures

The following grant information was disclosed by the authors:
AlMaarefa University.
Deanship of Scientific Research, Prince Sattam bin Abdulaziz University: PSAU/2023/R/1444.
Deanship of Scientific Research, Vice Presidency for Graduate Studies and Scientific Research, King Faisal University, Saudi Arabia: 2740.

### Competing Interests

The authors declare that they have no competing interests.

### Author Contributions

- Shtwai Alsubai conceived and designed the experiments, analyzed the data, prepared figures and/or tables, and approved the final draft.
- Ashit Kumar Dutta conceived and designed the experiments, performed the experiments, performed the computation work, prepared figures and/or tables, and approved the final draft.
- Abdullah M. Alnajim performed the experiments, analyzed the data, prepared figures and/or tables, authored or reviewed drafts of the article, and approved the final draft.
- Abdul rahaman Wahab sait conceived and designed the experiments, analyzed the data, performed the computation work, prepared figures and/or tables, and approved the final draft.
- Rashid Ayub conceived and designed the experiments, prepared figures and/or tables, authored or reviewed drafts of the article, and approved the final draft.
- Afnan Mushabbab AlShehri performed the experiments, authored or reviewed drafts of the article, and approved the final draft.
- Naved Ahmad performed the experiments, prepared figures and/or tables, and approved the final draft.

### Data Availability

The IoT Malware Dataset is available at Kaggle: https://www.kaggle.com/anaselmasry/iot-malware. The dataset is owned by Anas Aabo.

The Virusshare dataset is available at VirusShare: https://virusshare.com. The dataset is owned by Fosezo Cazade.

## Supplemental Information

Supplemental information for this article can be found online at http://dx.doi.org/10.7717/peerj-cs.1366#supplemental-information.

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
