# Peer review of "Artificial intelligence-driven malware detection framework for internet of things environment"

_PeerJ Computer Science, doi:10.7717/peerj-cs.1366_

## Round 0.1 · original submission · Major Revisions

Dear Authors,
Please revise and resubmit your manuscript. Clearly specify the motivation of this work. Reviewer 2 has requested that you cite specific references. You may add them if you believe they are especially relevant. However, I do not expect you to include these citations, and if you do not include them, this will not influence my decision.

Thank you.

·

Basic reporting

The paper “Artificial Intelligence-Driven Malware Detection Framework For Internet of Things Environment (#81530)” proposes a malware detection framework for IoT networks. The framework is tested with two datasets, namely IoT malware and Virusshare. The evaluation metrics used in the paper are accuracy, precision, recall, F1-score, Matthews correlation coefficient, and Kappa.

The related works are good and comprehensive. However, authors should represent the related work in table form with details (publish year, datasets, methods/algorithms, results, platforms, etc.)

The figures provided in this paper have to be improved in quality. Also, tables and figures need to be placed properly on the paper.

The objectives/contributions (i and ii) are unclear. Authors should explain what is new (improved) in this paper compared to relevant works.

The IoT malware dataset has no description of how input images are created.

Some references use links on the paper that need to be fixed. For example, Elmasry dataset ( https://www.kaggle.com/anaselmasry/iot-malware, (accessed Aug. 08, 2021))

An algorithm should be presented as an algorithm, not as a figure.

Overall, the framework is proposed for IoT networks. However, the paper has not pointed out where in an IoT network to apply the framework. The hardware platform used in this paper is software-based with CPU and GPU, which are computationally expensive.

Experimental design

Training configuration is not mentioned.

Authors should insert numbers/data in Figures 6 and 7. Line charts could be a better choice in this case.

Validity of the findings

no comment

Additional comments

Insert spaces when mentioning related works. For example, virtual worlds(Mu et al.,
2021; Atitallah, Driss & Almomani, 2022). → virtual worlds (Mu et al.,
2021; Atitallah, Driss & Almomani, 2022).

In the abstract: "Due to limited computing resources, conventional MD approaches failed to protect IoT devices" -> This sentence is incorrect; authors should move it to the introduction section, make it clear with proof and provide references.

In the abstract: “The framework generates the outcome at an accuracy, precision, recall, F1-score, Matthews correlation coefficient, Kappa of 98.65,98.7,98.3,98.5,97.5, and 97.65 and 97.3,96.4,96.85,96.63,95.25,and94.85 for IoT malware and Virusshare datasets, respectively.” → No need to show all the results in the abstract. Maybe just accuracy and F1-score?

Insert “)” in Equation 1

Reviewer 2 ·

Basic reporting

This article provides relatively comprehensive evaluations in various metrics, especially comparing the proposed model with existing models. Overall, the article is well organized, innovative and interesting at the same time.

Experimental design

- The authors should explain why DL technique is used rather than classic ML techniques (such as shallow neural nets).
- How are the hyper parameters used in the work chosen? Is it random or did the authors used any parameter tuning method?
- Did you noticed or expect computation overhead, or minor responsiveness because of the execution of the proposed algorithm?
- Intuitively describe why the proposed technique results in better performance?
- Authors compared the proposed model with existing models regarding accuracy, precision, and recall. However, the authors only described the results and did not analyze why the proposed model can be better than the existing models? Are the contributions from the deep learning model or something you improved in this work?
Where is the proposed network model? How can the proposed IDS detect the intrusions, provide a network model. Check the following articles.
1- https://doi.org/10.3390/electronics10080918
2- https://doi.org/10.3390/s21144884
3- doi: 10.1109/ACCESS.2022.3172304

The IoT devices are already resource constrained devices. The proposed IDS will put extra burden on such devices. How can the proposed IDS is efficient?
The authors used CNN, the CNN is mostly used for image classification. Why the authors used CNN based model. Further, it also requires high computational power.

Validity of the findings

There have been numerous works recetly on IDS. How is tis article different from them?
What are the threats to validity of the proposed model?

Additional comments

As additional comments: I suggest the authors to proofread the entire manuscript as it is having lot of grammatical mistakes.

Reviewer 3 ·

Basic reporting

The authors seek to present an Artificial Intelligence-Driven Malware Detection Framework for the Internet of Things Environment. Overall, the authors made significant efforts. However, I would suggest a few recommendations.
1. A clear motivation is missing in the paper. I would suggest the authors add a detailed motivation for their study to provide context for their research.
2. Instead of listing objectives, I would suggest the authors list their novel contributions to AI-based malware detection. This will provide a clearer picture of the significance of their work.
3. The entire literature is written in the present tense, which is not ideal for a literature review. I would suggest the authors change their words to the past tense. For example, change "proposed" to "propose," "presented", to "present," and so on. This will help convey that they are referring to previous work.

Experimental design

1. The section on the Materials and Methods is very well organized, and it gives a very clear explanation of the various methods and techniques that were utilized in the study. It would be helpful to include more specifics on the dataset that was used as well as the pre-processing steps that were carried out in order to get the data ready. Besides, algorithm cannot be written in image form.
2. In the results section, a comprehensive analysis of the performance of the AI-based malware detection for IoT is presented. However, it needs improvement. Specifically, there is a need for additional explanation regarding the figures and Tables because it is possible that a few lines will not be sufficient to understand the results fully. Additionally, there is a need for an improvement in the quality of the Figures, as all of the Figures included in the paper are currently quite blurry.
3. In the section titled "Discussion," insightful commentary is provided on the results and the implications of those results. It would be helpful to address any possible limitations of the study as well as identify areas that need further investigation. Besides, the section needs a more detailed overview.

Validity of the findings

The findings in the paper are impressive, but the results section could use more explanation. The tables and figures need clearer descriptions so that readers can understand them better

Additional comments

1. The references are not enough, and the authors need to add more references. Additionally, the authors have ignored some of the latest papers in the current domain, and a comparative analysis with these papers would be beneficial.
2. In general, the research paper offers insightful information and I would suggest a major revision.

---

## Round 0.2 · accepted · Accept

The authors have addressed all of the reviewers' comments.

Reviewer 2 ·

Basic reporting

The authors have addressed all of my concerns. I have no further comments.

Experimental design

NA

Validity of the findings

NA

Additional comments

NA

Reviewer 3 ·

Basic reporting

The authors have addressed my comments, I have no further comments. I believe the paper is in good shape and can be published now.

Experimental design

I believe it's improved.

Validity of the findings

The findings are enough.